# The New Quadrivalent Adjuvanted Influenza Vaccine for the Italian Elderly: A Health Technology Assessment

**DOI:** 10.3390/ijerph19074166

**Published:** 2022-03-31

**Authors:** Giovanna Elisa Calabrò, Sara Boccalini, Donatella Panatto, Caterina Rizzo, Maria Luisa Di Pietro, Fasika Molla Abreha, Marco Ajelli, Daniela Amicizia, Angela Bechini, Irene Giacchetta, Piero Luigi Lai, Stefano Merler, Chiara Primieri, Filippo Trentini, Sara Violi, Paolo Bonanni, Chiara de Waure

**Affiliations:** 1Section of Hygiene, University Department of Life Sciences and Public Health, Università Cattolica del Sacro Cuore, 00168 Rome, Italy; marialuisa.dipietro@unicatt.it; 2VIHTALI (Value in Health Technology and Academy for Leadership & Innovation), Spin Off of Università Cattolica del Sacro Cuore, 00168 Rome, Italy; 3Department of Health Sciences, University of Florence, 50121 Florence, Italy; sara.boccalini@unifi.it (S.B.); angela.bechini@unifi.it (A.B.); paolo.bonanni@unifi.it (P.B.); 4Department of Health Sciences, University of Genoa, 16132 Genoa, Italy; donatella.panatto@unige.it (D.P.); daniela.amicizia@unige.it (D.A.); pierolai@unige.it (P.L.L.); 5Clinical Pathways and Epidemiology Unit-Medical Direction, Bambino Gesù Children’s Hospital, IRCCS, 00165 Rome, Italy; caterina1.rizzo@opbg.net; 6Graduate School of Health Economics and Management, Università Cattolica del Sacro Cuore, 00168 Rome, Italy; abreha.fasikamolla01@icatt.it; 7Laboratory for Computational Epidemiology and Public Health, Department of Epidemiology and Biostatistics, Indiana University School of Public Health, Bloomington, IN 47405, USA; majelli@iu.edu; 8Department of Medicine and Surgery, University of Perugia, 06123 Perugia, Italy; irene.giacchetta@studenti.unipg.it (I.G.); chiara.primieri@studenti.unipg.it (C.P.); sara.violi@studenti.unipg.it (S.V.); chiara.dewaure@unipg.it (C.d.W.); 9Center for Health Emergencies, Bruno Kessler Foundation, 38122 Trento, Italy; merler@fbk.eu (S.M.); filippo.trentini@unibocconi.it (F.T.); 10Dondena Centre for Research on Social Dynamics and Public Policy, Bocconi University, 20136 Milan, Italy

**Keywords:** influenza, vaccination, vaccines, Health Technology Assessment, HTA, quadrivalent adjuvanted influenza vaccine, elderly

## Abstract

Background. The elderly, commonly defined as subjects aged ≥65 years, are among the at-risk subjects recommended for annual influenza vaccination in European countries. Currently, two new vaccines are available for this population: the MF59-adjuvanted quadrivalent influenza vaccine (aQIV) and the high-dose quadrivalent influenza vaccine (hdQIV). Their multidimensional assessment might maximize the results in terms of achievable health benefits. Therefore, we carried out a Health Technology Assessment (HTA) of the aQIV by adopting a multidisciplinary policy-oriented approach to evaluate clinical, economic, organizational, and ethical implications for the Italian elderly. Methods. A HTA was conducted in 2020 to analyze influenza burden; characteristics, efficacy, and safety of aQIV and other available vaccines for the elderly; cost-effectiveness of aQIV; and related organizational and ethical implications. Comprehensive literature reviews/analyses were performed, and a transmission model was developed in order to address the above issues. Results. In Italy, the influenza burden on the elderly is high and from 77.7% to 96.1% of influenza-related deaths occur in the elderly. All available vaccines are effective and safe; however, aQIV, such as the adjuvanted trivalent influenza vaccine (aTIV), has proved more immunogenic and effective in the elderly. From the third payer’s perspective, but also from the societal one, the use of aQIV in comparison with egg-based standard QIV (eQIV) in the elderly population is cost-effective. The appropriateness of the use of available vaccines as well as citizens’ knowledge and attitudes remain a challenge for a successful vaccination campaign. Conclusions. The results of this project provide decision-makers with important evidence on the aQIV and support with scientific evidence on the appropriate use of vaccines in the elderly.

## 1. Introduction

Influenza is a serious public health problem and has a significant epidemiological, clinical, and economic impact on health systems worldwide [1]. The World Health Organization (WHO) estimates that worldwide annual influenza epidemics result in about 3 to 5 million severe cases of illness, especially among the elderly, younger children (<5 years of age), pregnant women, and individuals with chronic diseases [2]. Lower and upper respiratory tract infections are the main consequences of seasonal influenza. It was estimated that approximately 290,000–650,000 deaths from respiratory causes are attributable to influenza each year [3]. In addition, several extra-respiratory complications, such as those of the cardiovascular and nervous systems, have an important impact, especially on the most vulnerable patients. In high-income countries, most influenza-related deaths occur among adults aged 65 and over [3]. Recent evidence shows the significant burden that influenza places each year on the Italian population across all age groups. In particular, a significantly increased risk for complications is shown among the elderly (65+) and patients with at least one chronic condition. With respect to mortality, influenza is responsible for a relevant excess in mortality rate, in particular in persons 65+ [3].

Therefore, influenza-related morbidity and mortality are greatest among the elderly, commonly defined as subjects aged ≥65 years [4]. However, vaccine immunogenicity, efficacy, and effectiveness are the lowest in this vulnerable population, mainly due to the decrease in the immune response with age, a phenomenon known as “immunity senescence” [4]. In order to improve immunogenicity, vaccine efficacy, and effectiveness in the elderly population, various lines of research were investigated over the years, such as the use of adjuvant systems, the increase in the antigen dose, the administration of a booster dose, and intradermal and virosomal vaccines. The two most promising lines of research led to the authorization of vaccines specifically designed for the over 65 population, namely the MF59-adjuvanted quadrivalent influenza vaccine (aQIV) and the high-dose quadrivalent influenza vaccine (hdQIV) [4]. The elderly are among at-risk subjects for whom annual influenza vaccination is recommended in European countries [5], and possible strategies for dealing with it include adjuvanted and high doses vaccines and herd immunity induced by increased vaccine uptake in all age groups [6].

Today, healthcare systems are constantly in search of effective strategies in the context of prevention: among all, vaccination certainly occupies a role of absolute pre-eminence [7]. Therefore, new vaccines must be evaluated in order to maximize the results in terms of achievable health benefits, guarantee the community adequate protection, and support evidence-based decision-making. At the international level, Health Technology Assessment (HTA) is meant to determine the value of health technology and then inform decision-making [8].

For this reason, we carried out an HTA of the new aQIV by adopting a multidisciplinary policy-oriented approach to evaluate the clinical, economic, organizational, and ethical implications for the Italian elderly.

## 2. Materials and Methods

The report was developed according to the Core Model^®^ EUnetHTA [9] that includes the assessment of nine domains such as (1) health problems and current use of the technology, (2) description and technical characteristics of technology, (3) safety, (4) clinical effectiveness, (5) costs and economic evaluation, (6) ethical analysis, (7) organizational aspects, (8) patients and social aspects, (9) and legal aspects.

In our report, we included both clinical (the health problem—epidemiology and burden of influenza, characteristics, immunogenicity/effectiveness, and safety of available vaccines) and non-clinical (economic evaluation of aQIV and organizational, ethical, and social implications of influenza vaccination) domains.

The Italian elderly, defined as subjects aged ≥65 years, represented the study population.

The full HTA report was published in the Italian language in May 2021 [10].

### 2.1. Clinical Domains

The health problem was analyzed through both a systematic review of the available Italian literature and the consultation of national health statistics from InfluNet, FluNews, and Health for All. Particularly, Influenza-Like Illness (ILI) attack rates and data from the virological surveillance in nine influenza seasons (2010/2011–2018/2019) were collected and analyzed; the atypical epidemic seasons 2009/2010 and 2019/2020 were excluded from the analysis. The disease burden was evaluated in terms of influenza-related deaths, hospitalizations for influenza and pneumonia, and complications. The methods of the systematic review were reported in detail in another paper [3].

The features of aQIV were collected from the Summary of Product Characteristic (SmPC) [11], whereas its immunogenicity, efficacy, and safety were evaluated by looking for both papers published on PubMed until 19 November 2020, and assessment reports and clinical reviews available on the European Medicines Agency (EMA) and Food and Drug Administration (FDA) websites. Furthermore, because data on the adjuvanted trivalent influenza vaccine (aTIV) are considered transferable to the aQIV, an update of the already published systematic reviews and meta-analyses of aTIV was performed using the same original search syntax and methodology [12]. 

The egg-based standard quadrivalent vaccine (eQIV) was considered the alternative. Therefore, research of published systematic reviews and meta-analyses on its immunogenicity, efficacy, effectiveness, and safety was performed on PubMed, Embase, and Cochrane Library. In the absence of systematic reviews, Randomized Clinical Trial (RCT) of eQIVs were considered for immunogenicity and efficacy outcomes from the influenza season 2007/08 (when the first trial was conducted), whereas, regarding effectiveness, studies carried out since the influenza season 2017/18 season were evaluated.

### 2.2. Non-Clinical Domains

A deterministic transmission model was already published [13] and used to simulate the population-level dynamics of influenza infection (Box 1). 

Box 1Description of the Transmission Model.Influenza transmission was simulated using a deterministic Susceptible-Exposed-Infectious-Removed (SEIR) model to calculate attack rates of seasonal influenza confirmed infection by age group and viral subtype over the seasons considered. The model was stratified into 85 age groups and is based on the assumption of a heterogeneous mixing for considered each age group.The susceptibility and immunity of the Italian population to infection was defined using the data on strains circulating in the last 10 seasons (from 2010/2011 to 2019/2020, excluding the 2009/2010 pandemic season) and starting from sero-epidemiological investigations carried out before and after the appearance of A(H1N1)pdm09 subtype in Italy [14,15].The model was calibrated on the data of laboratory-confirmed influenza cases in Italy and on the number of ILI reported to the Sentinel Surveillance System for influenza-like syndromes (InfluNet) by sentinel doctors (general practitioners and pediatricians) participating in surveillance [16].

The economic evaluation used a deterministic (DSA) and probabilistic sensitivity analysis (PSA) approach. The results of the transmission model served as inputs for the economic DSA and PSA in which clinical and economic parameters were incorporated, and outcomes were compared for different scenarios. In the basic scenario, we considered the vaccination with the eQIV from six months upwards, while in the alternative scenario, we considered the introduction of the aQIV in the whole population over 65 years old. Regarding the new hdQIV, indicated for subjects aged ≥65 years [17], no formal cost-effectiveness analysis was conducted to compare it with aQIV. This choice was determined by the fact that the studies available so far on the relative effectiveness of aQIV vs. hdQIV [18,19,20,21,22] use poorly specific influenza-related outcomes (i.e., no study used laboratory-confirmed influenza as the primary outcome) and cannot be considered conclusive. Elsewhere, any non-vaccination strategy was not taken into consideration as it was deemed implausible as influenza vaccination is highly recommended by all national and international health authorities. Vaccination coverages (CVs) were extracted by official institutional reports [23]. Because in individuals with comorbidities, the probability of being vaccinated is higher, but CVs among risk categories in individuals ≥18 years were not available. In order to estimate the CVs in those at higher risk of complications due to influenza, we applied a correction based on the relative risk (RR) of being vaccinated having at least one underlying chronic disease using the data of “Passi” [24] and “Passi d’argento” [25] surveillance systems. We estimated a RR (high vs. low) of being vaccinated of 4.2 and 1.4 for subjects aged 18–64 and ≥65 years, respectively. Finally, since the 2020/21 season was characterized by a significant increase in the demand for influenza vaccines, we conducted a scenario analysis where the CVs were increased by 25% compared to those available from official reports referring to 2019/20. 

As for the other data inputs, the probability of symptomatic infection and of a subsequent visit to the general practitioner or to the emergency room by age group was obtained from the scientific literature [26,27,28]. The hospitalization rates by age group due to influenza were collected from the same studies [27,28]. Influenza-related mortality was estimated using the fraction of deaths from all causes associated with influenza [29]. All the above estimates were drawn from national data. All the remaining probabilities relating to the natural history of influenza were drawn from the international scientific literature [10,30].

The direct and indirect costs were extrapolated and/or reconstructed starting from the official rates and/or from ad hoc Italian studies. Included costs were related to (i) vaccination campaign, (ii) management of symptomatic influenza cases, and any consequent events such as complications. All the parameters used to reconstruct the direct and indirect costs relating to influenza management are reported in the full HTA report [10].

As recommended by the WHO [31] and the Consolidated Health Economic Evaluation Reporting Standards (CHEERS) guidelines [32], the analysis was conducted both from the perspective of the NHS (direct costs) and from the broader perspective of society (direct and indirect costs). Costs were inflated to 2020, and a 3% discount rate of QALY and indirect costs for adverted deaths was considered [33].

The intervention (aQIV vaccination in all elderly) was considered cost-effective for a value of ICER < €30,000/QALY in both perspectives considered.

Concerning the organizational aspects, current national recommendations [34] on influenza prevention and control, issued in recent years [17,35,36,37], were consulted. Particularly, the indications of any preferential use of available influenza vaccines in the elderly were researched. In addition, a narrative review of the international literature was carried out to identify available evidence to support the value of the appropriateness of the use of different influenza vaccines in the elderly.

Furthermore, based also on the definition of “Value-Based Health Care” (VBHC) proposed by the Expert Panel on Effective Ways of Investing in Health (EXPH) of the European Commission (EC) [38], and in line with the agenda promoted in September 2019 by the EC and the WHO [39], the issue of the individual and social value of influenza vaccination was addressed. In particular, in order to better understand the attitude of the general population towards vaccines and vaccination, an overview of the main European and Italian documents [40,41,42,43,44] on knowledge, attitudes, opinions, and perceptions of citizens was conducted.

Finally, in order to assess any ethical issue raised by the implementation of aQIV within the Italian NHS, we used the so-called “triangular method” [45] that places the value and the respect of humans at the center of the ethical reflection. In light of this approach, we analyzed the vaccine risk/benefit ratio, quality of life improvement, exercise of autonomy, cost-effectiveness ratio, prioritization of healthcare policies, and equal access to the vaccination.

## 3. Results

### 3.1. Clinical Domains

In Italy, the elderly have a lower cumulative incidence (4.3%) of ILI compared to other age groups, but this corresponds to about 600,000 subjects suffering from ILI every year (reaching one million in seasons with greater intensity). These cases are predominantly distributed within a few weeks (on average, seven weeks), reaching a weekly number ranging from 25,000 to 120,000 cases to be managed at the same time in a short period [46].

From the virological point of view, in the last years, there has been a greater diffusion of influenza viruses A. The seasons with a predominant circulation of virus A(H3N2) were associated with an excess of mortality from pneumonia and influenza for subjects ≥65 years old [47,48] compared to seasons dominated by A(H1N1) or B.

The number of severe cases requiring hospitalization in the intensive care unit or extracorporeal membrane oxygenation stratified by age groups was available only for the 2017/2018 and 2018/2019 influenza seasons and showed that 39–45% of cases were elderly [28]. Interestingly, the proportion of severe cases due to influenza B strain was very high in the elderly as compared to other age groups in the 2017/2018 influenza season. Hospitalization rates for influenza and pneumonia were higher in males than females (90.19/10,000 and 50.88/10,000, respectively, in 2018), and people aged 75 years or older showed rates up to three times higher than elderly aged 65–74 years. This last datum was also confirmed for mortality rates for influenza and pneumonia. According to the national evidence [28,46], 77.7% to 96.1% of influenza-related deaths occur in the elderly.

The immunogenicity, efficacy, and safety of aQIV were evaluated by two studies, the V118_20 [49] and the V118_18 [50].

The V118_20 study [49] was a multicentre randomized controlled double-blinded study performed in the 2017/2018 influenza season, which enrolled 1778 subjects ≥65 years old. The study demonstrated that aQIVs were non-inferior to aTIVs with regard to the shared strains and superior in respect to the alternative strain of aTIV. These results were not influenced by age and comorbidities. The safety profile of the two vaccines was similar, with most of the adverse events being mild and short-term.

Beran et al. [50] published the results of V118_18 after the closure of our literature search on PubMed. Nevertheless, the results of V118_18 were already available from the regulatory agencies’ websites. The V118_18 study was a randomized, stratified, observer-blind, controlled, multicentre, phase 3 study carried out in the 2016–17 northern hemisphere and 2017 southern hemisphere influenza seasons. The primary aim was to investigate the absolute efficacy of aQIV with respect to laboratory-confirmed influenza. The study failed to achieve the primary endpoint, probably due to the circumstances related to the mismatch between the influenza strain included in the vaccine and the virus A(H3N2) circulating in the study period.

Relating to aTIV, our updated systematic review and meta-analysis [10] showed that aTIV was more immunogenic than egg-based standard trivalent influenza vaccine (eTIV) versus both homologous and drifted strains; of note, the advantage of aTIV vs. eTIV was higher (+35%) regarding to the drifted A(H3N2) strains. The meta-analytic estimates reported that the magnitude of the antibody response was 26–51% higher in subjects vaccinated with aTIV than for vaccines with eTIV. The absolute effectiveness of aTIV was generally significantly different from zero, independently of setting and influenza season; it resulted in 44% (95% CI: 21–60%) [10]. Eventually, aTIV was generally highly effective compared to eTIV or eQIV independently by considered outcome; specifically pooled relative effectiveness was 34.6% (95% CI: 2.0–66.0%; *p* < 0.05) in preventing laboratory-confirmed influenza.

Regarding safety and tolerability, when compared with non-adjuvanted vaccines, adjuvanted vaccines may cause more solicited adverse events [10]. Specifically, aTIV was found to be more reactogenic at the injection site (89%, 54%, and 58% for pain, erythema, and induration, respectively). By considering the systemic reactions, a significant increase in chills (RR = 1.48) and myalgia (RR = 1.39) was observed. Instead, the increase in the frequency of fever, general malaise, arthralgia, and headaches was not significant [10].

Regarding eQIVs, no efficacy studies have been published so far in the adult and elderly population, as well as any systematic review and meta-analysis on the effectiveness in the elderly was found. Three systematic reviews/meta-analyses were included with respect to immunogenicity. Specifically, the first [51] published in 2016 reported the non-inferiority of eQIV vs. eTIV for shared strains and the superiority of eQIV for the fourth B strain, which was not included in eTIVs. The second review [30] published in 2019 considered the absolute immunogenicity of eQIV in the elderly. In all studies, the immune response versus A(H1N1), A(H3N2), B(Victoria), and B(Yamagata) homologous strains satisfied the European CHMP (Committee for Medicinal Products for Human Use) and US CBER (Center for Biologics Evaluation and Research) criteria. The third review published in 2020 [52] found that the SeroProtection Rate and the SeroConversion Rate of older adults (≥60 years old) were lower than younger adults for A(H1N1) and B(Victoria), while the two age groups had similar antibody responses for A(H3N2).

The studies on effectiveness found that the absolute effectiveness of eQIVs changes in relation to influenza season, age group, and study outcome (e.g., laboratory-confirmed influenza, hospitalization for influenza/pneumonia, etc.) and showed a decreasing trend with increasing age [53,54,55].

### 3.2. Non-Clinical Domains 

The economic evaluation showed that in an average season, the introduction of aQIV could avoid 111,417 (95%CI: 61,803–201,338) influenza cases (both symptomatic and asymptomatic) (Figure 1). The hospitalizations and deaths avoided would be 363 (95% CI: 198–658) and 195 (95% CI: 106–353), respectively. In particular, 93% of avoided hospitalizations and 98% of avoided deaths would be recorded in those over 65 years old. From the NHS perspective, the aQIV strategy is cost-effective (ICER: EUR 12.556/QALY) for the whole population considered. When considering only subjects aged ≥65 years, it is cost-effective (ICER: EUR 14.441/QALY), while it is dominant in the population not vaccinated with aQIV (6 months-64 years) due to an indirect effect. As expected, no significant difference was found in the outcome of the two perspectives (NHS and societal perspectives); indeed, from the societal perspective, the aQIV strategy is cost-effective, with a slightly lower ICER (EUR 11.748/QALY) (Table 1).

The DSA demonstrated that the main drivers of the ICER (regardless of the study perspective) were the total number of infections (and therefore vaccine efficacy); purchase price of both aQIV and eQIV; and probability of help-seeking, death, and complications. The PSA showed that the aQIV strategy is cost-effective in over 95% of the simulations (Figure 2).

With increasing vaccination coverage by 25%, the number of events avoided further increases. In particular, the introduction of aQIV in the Italian elderly could avoid 126,234 (95% CI: 70,929–225,499) influenza cases (both symptomatic and asymptomatic), while the number of hospitalizations and deaths avoided would be, respectively, 414 (95% CI: 227–737) and 223 (95% CI: 122–395).

In the National Immunization Plan 2017–2019 [31], which is still in force in Italy, influenza vaccination is recommended for individuals over the age of 64 years regardless of any risk situation. However, the plan does not report which vaccine must be preferred among available ones. In the Italian Health Ministry Circulars for influenza prevention and control for the 2018/2019 and 2019/2020 seasons, it was specified that the aTIV provides higher protection than non-adjuvanted trivalent and quadrivalent vaccines in older people (75 years or older) [35,36]. In the Circular for the 2020/2021 influenza season, instead, the indication for the elderly was more general: adjuvanted and non-adjuvanted vaccines, as well as high-dose and cell-based vaccines, are all recommended for people aged ≥65 years, without any specific indication of use [7]. In the last Circulation for the 2021/2022 season, instead, adjuvanted or high-dose QIV was reported as the most indicated vaccines in subjects ≥65 years of age [37]. Actually, the available scientific evidence shows that influenza vaccines for the elderly have not all equal effectiveness and cost-effectiveness. Therefore, they should be offered appropriately based on the vaccine’s characteristics, first of all, age. In fact, the use of the most appropriate vaccine could optimize the health benefits and reduce costs for the NHS [56]. 

With respect to citizens’ knowledge, attitudes, opinions, and perceptions, the document “State of Vaccine Confidence in the EU 2018” [40] report that, in the European Member States, the perception of the general population towards vaccines is positive, with the majority of citizens agreeing that vaccines are important (90.0%), safe (82.8%), and effective (87.8%). In particular, the Italian population agrees on the importance of vaccines (93.0%), their safety (94.0%), and effectiveness (94.0%). Most of the European Union (EU) population also agrees that the seasonal influenza vaccine is important (65.2%) and safe (69.4%). In Italy, there is also agreement on the importance (96.0%) and safety (96.0%) of flu vaccination. In addition, young people between 18 and 24 years of age and those 65 years old or older agree more than other age groups (between 25 and 64 years of age) on the importance of influenza vaccination. Each area of confidence and each vaccine addressed in the survey triggered different results across countries, showing how political and media discourse can shape a country’s confidence in the importance, efficacy, and safety of vaccines, including seasonal influenza. The survey also shows that confidence varies for different vaccines, highlighting the need for targeted responses to rebuild trust.

According to the 2019 Eurobarometer survey [41], only slightly more than half of Europeans (56% of respondents) are aware that influenza still causes deaths in the EU, with differences across countries. In Italy, only 15% of respondents are aware of the correlation between influenza and mortality. Furthermore, important information emerges with respect to the social value of vaccinations: 88% of Europeans express a certain level of agreement on the importance of vaccination to protect not only themselves but also others, and 87% agree that vaccination is important as it protects those who cannot be vaccinated. In Italy, 79% of the interviewees agree with both statements.

Eventually, from an ethical point of view, collected data allowed to release a positive judgment as aQIV induces an excellent antibody response against the four influenza strains in the elderly, it has a good safety profile, is cost-effective, and is not contrary to the values of human beings. 

## 4. Discussion

The results of this HTA project provide decision-makers with important evidence on the new aQIV. In fact, the assessment of new technology is an important requisite for a proper allocation of resources at a central level and for fostering its appropriate use at the local level. 

Beyond an updated overview of the evidence on the health problem, namely, influenza, based on national data, the HTA provides information on the efficacy and cost-effectiveness of aQIV, showing that it is as immunogenic as aTIV and cost-effective if compared to eQIV. These results line up with the other evidence regarding the previous aTIV in comparison to non-adjuvanted TIV [57,58,59]. 

As mentioned, such kind of evidence is also important to guide the appropriate use of available influenza vaccines. In this respect, it is noteworthy that for the 2021/2022 influenza seasons, several advisory committees, such as the USA Advisory Committee on Immunization Practices [60] and the Canadian National Advisory Committee on Immunization (NACI) [61], did not recommend the use of one vaccine over the other. On the other hand, the UK Joint Committee on Vaccination and Immunisation (JCVI) [62] recommended, for adults aged ≥65 years, the aQIV and hdQIV; instead, the Australian Technical Advisory Group on Immunisation (ATAGI) [63] preferably recommended adjuvanted influenza vaccines in the same age group. As a matter of fact, data on the relative effectiveness of the new influenza vaccines for the elderly, i.e., aQIV and hdQIV, in comparison to standard-dose unadjuvanted QIV, are limited [64]. Furthermore, data on the relative effectiveness of aQIV vs. hdQIV are also inconclusive [65] because of their recent introduction. Some evidence does exist with respect to their corresponding trivalent formulations (aTIV and hdTIV). 

People aged ≥65 years actually account for 23% of the Italian population [66], and they will increase in the future, reaching up to 34% in 2050 [67]. Furthermore, 61% of the elderly have at least one chronic disease, and 25% have two or more chronic diseases [68]. Furthermore, with the increase in life expectancy, the prevalence of chronic diseases is also projected to grow [68]. All these things together call attention to enhancing the appropriate use of available vaccines. 

It is desirable that influenza vaccination campaigns become more and more precise and adapted to subjects’ characteristics. That implies prioritizing the use of available vaccines in the light of allowing timely and appropriate access to them and pursuing value for money and universal health coverage [69]. The establishment of a formal link between the evidence issued by HTA and coverage and reimbursement decisions is therefore crucial [69]. This draws attention to the first criticism of HTA in Italy: the lack of a national agency entrusted to perform HTA and issuing recommendations. Furthermore, it should be pointed out that the procurement of influenza vaccines is tender-based and commissioned to Regional Health Authorities with high heterogeneity in procured types and brands [70,71]. The development of special guidelines on the use of available influenza vaccines could help overcome these regional differences. Even though this task is not a priority for HTA [69], in some countries, such as UK or Australia, HTA bodies are also committed to producing or validating guidelines.

At the European level, the harmonization of HTA has come to fulfillment with the approval of the HTA regulation on HTA on 13 December 2021 [72]. The new regulation is aimed at improving the sustainability of public health systems and improving patients’ access to medicines. It envisages the development of a Joint Clinical Assessment (JCA) that Member States should consider in their national HTA. Indeed, the regulation moves in the direction of “globalize the evidence, localize the decision” [73], laying the foundation of shared information at the European level. Shared information, together with early engagement with vaccine stakeholders and improvement of transparency of the decision-making process, was also recommended for improving vaccine market access [70]. Nevertheless, because of vaccines uniqueness, Vaccines Europe, a specialized vaccines group within the European Federation of Pharmaceutical Industries and Associations, pinpointed some aspects that should be paid attention to in performing future JCA, namely the involvement of public health and vaccines experts as well as vaccines industries in the assessment and the development of specific methodologies and processes for JCA on vaccines [74]. The latter should take into consideration the broad value of vaccines, namely the short- and long-term benefits that fall outside the clinical setting [75,76]. The HTA approach could help to disentangle and appraise this broad value [75,76].

Alongside, innovative and evidence-based effective vaccination strategies should be identified to increase vaccination uptake in all age groups [77], especially the elderly. In this respect, the COVID-19 pandemic highlighted that it is necessary to be prepared to appropriately use new and old weapons to protect people and reduce the pressure on health systems [78].

Our study presents some limitations. The most meaningful one was the choice not to compare aQIV with hdQIV, but, as mentioned above, the available literature does not provide conclusive evidence about their relative effectiveness. Nevertheless, a recent economic analysis aimed at comparing aQIV with hdQIV concluded that aQIV could be considered cost-saving unless hdQIV is not priced lower than hdTIV [79]. In order to acquire new knowledge, further investigation should be performed to assess the relative effectiveness and the value of these new vaccines. Other limits of the study were represented by the need to use international evidence to gather some inputs for the economic evaluation. This was due to the lack of national data that should be further and continuously issued to make it possible for a more thorough, timely, and context-based picture of the whole burden of influenza in Italy [59]. Our research contributes to the production of evidence to develop vaccination policy recommendations based on a robust, multidisciplinary, and multidimensional approach, highlighting the opportunity and the appropriateness of using new vaccines to enhance the value of preventive interventions in public health. Nonetheless, decision-makers should pay particular attention to implementing health education campaigns targeting both the population and healthcare professionals, guarantee equal access to vaccination, and adequately monitor the effectiveness and safety of aQIV in post-marketing studies.

## 5. Conclusions

Currently, a new vaccine (aQIV) is available for the immunization of the elderly. The evidence suggests that it is as immunogenic as aTIV and that adjuvanted vaccines have a significant absolute and relative effectiveness compared to egg-based standard vaccines. The multidisciplinary policy-oriented assessment that was carried out in our HTA project showed that the use of aQIV in the elderly population is cost-effective and moves in the direction of strengthening the appropriateness of the use of available influenza vaccines. Evidence from our HTA could guide the appropriate use of available vaccines for a successful vaccination campaign and help reduce the burden of influenza on the elderly.

## Figures and Tables

**Figure 1 ijerph-19-04166-f001:**
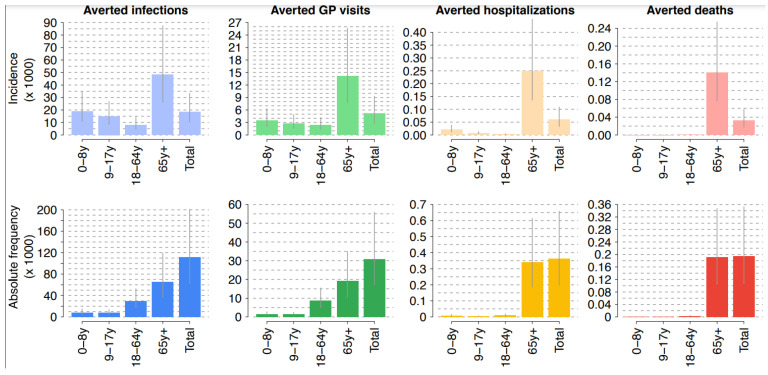
Influenza burden avoided thanks to the introduction of aQIV in Italy.

**Figure 2 ijerph-19-04166-f002:**
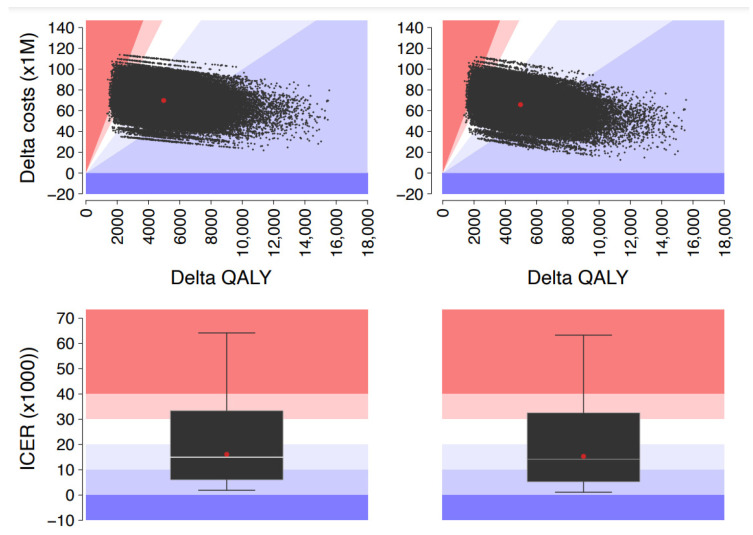
Probabilistic sensitivity analysis: base scenario (QIVe) vs. alternative scenario (aQIV in subjects aged ≥65 years), by study perspective (NHS on the **left**, society on the **right**).

**Table 1 ijerph-19-04166-t001:** Case base: cost-effectiveness comparison between base scenario (QIVe in all groups) and alternative scenario (aQIV in subjects aged ≥65 years), by study perspective and age group.

Age Group	Total Costs, €		Incremental Cost, €	QALY		Incremental QALY, ΔQALY	ICER, €/QALY
	Base Scenario	Alternative Scenario		Base Scenario	Alternative Scenario 1		
**National Health System (NHS) Perspective**							
0.5–8	74,022,455	73,743,288	−279,167	15,810	15,748	62	Dominant
9–17	45,665,939	45,473,834	−192,105	11,414	11,365	49	Dominant
18–64	168,215,572	166,978,400	−1,237,172	50,846	50,407	439	Dominant
≥65	147,357,045	213,374,855	66,017,809	41,046	36,474	4572	14,441
Total	**435,261,011**	**499,570,376**	**64,309,365**	**119,116**	**113,994**	**5122**	**12,556**
**Societal Perspective**							
0.5–8	104,431,875	104,033,764	−398,112	15,810	15,748	62	Dominant
9–17	58,791,065	58,542,301	−248,764	11,414	11,365	49	Dominant
18–64	626,765,954	621,568,549	−5,197,404	50,846	50,407	439	Dominant
≥65	147,357,045	213,374,855	66,017,809	41,046	36,474	4572	14,441
Total	**937,345,939**	**997,519,469**	**60,173,530**	**119,116**	**113,994**	**5122**	**11,748**

## Data Availability

Not applicable.

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
