# Peer review of "The New Quadrivalent Adjuvanted Influenza Vaccine for the Italian Elderly: A Health Technology Assessment"

_ijerph, 2022, doi:10.3390/ijerph19074166_

Round 1
Reviewer 1 Report
The concept of the study was potential for the scientific community; however, for improving the manuscript from the current stage, I have the following comments:
- Line 29 on page 1, the author may mention the age group of the elderly population.
- Line 41 on page 1, please spell out aTIV acronym.
- In the Introduction, I found a lack of flow of the background. The authors may start from the second paragraph. In Line 64, at the end of this paragraph, the authors may write more about the scenario of Italy besides the high-income countries. The authors may also drop the third paragraph. In Line 80, please add some more information related to the efficiency of the hdQIV vaccine.
- Line 102 on page 3, please write briefly why it was excluded.
- In section 3.1, it seems that the authors made a review of the selected articles. It would be better if they make a table and show the results in the place of long writing.
- Line 262 on page 6, is it 12.566
- Line 268 on page 6, is it 11.748
- Line 302-317 on page 8, the authors may write more about Italy than the EU.
- Line 318-324 on page 8, the authors may drop it.
- Line 325-335 on page 8, the authors may move it to the policy implication section at the end of the Discussion section.
Author Response
Dear Editor and Dear Reviewer,
We would like to thank you for the opportunity to resubmit our work. We have amended the paper according to the received suggestions and we hope that it now appears improved.
Hereafter point-to-point answers are provided.
Best regards,
Giovanna Elisa Calabrò

Reviewer 2 Report
In the manuscript titled “The new quadrivalent adjuvanted influenza vaccine for the Italian elderly: a health technology assessment” by Giovanna Elisa Calabrò et.al., the authors have carried out a Health Technology Assessment (HTA) of the new adjuvanted quadrivalent influenza vaccine (aQIV) by adopting a multidisciplinary policy-oriented approach to evaluate the clinical, economic, organizational and ethical implications for the Italian elderly.
This reviewer would like the authors to address the following points
(1) (Line 205-206) -- The authors have mentioned that "Hospitalization rates for influenza and pneumonia were higher in males than females (90.19/10,000 and 206 50.88/10,000, respectively, in 2018)……". This is a significant difference. The authors are requested to provide plausible reasons.
(2) Figure 1 -- There is a difference of a few folds in between averted infections and averted deaths (absolute frequency). Why is that?
(3) (Line 304-307) -- The authors have reported that 90% of the citizens agree that vaccines are important. In case of seasonal influenza vaccine, this percentage drops down to 65%. Why is that?
Other than that, this manuscript is a good fit for acceptance in IJERPH .
Author Response

(The authors gave the same response as above.)

Reviewer 3 Report
The manuscript with the title "The new quadrivalent adjuvanted influenza vaccine for the 2 Italian elderly: a health technology assessment" is well structured and provides information on the scientific evidence for the use of a vaccine available for immunization of the elderly.
Of the 76 references, 7 are over 10 years old, 4 are between 5-10 years old and 59 are from the last 5 years. I noticed 6 references that do not have access data: 23,24,44,66,72
References 9,11,12,13,14,17,18,19,20,21,25,26,29,31,46,47,48,51,52,53,57,58,74,75 not all the authors of the works are listed, which means that in these references we can also find the names of the authors of this article. In these references it appear the names of the authors of this article for several works:
-7 times- Rizzo C
-5 times- Calabrò GA
-4 times -Boccalini S, Bechini A, by Waure C
-3 times - Bonanni P
-one time - Panatto D, Di Pietro ML, Ajelli M, Amicizia D, Giacchetta I, Lai PL, Primieri C, Trentini F
The data included in the manuscript is based on scientific evidence
The results presented in the manuscript can be a starting point for further research and for other countries.
Minor comments:
Introduction
The manuscript entitled "The new quadrivalent adjuvanted influenza vaccine for the 2 Italian elderly: a health technology assessment" presents data on the use of an influenza vaccine in the elderly, data supported by scientific evidence. The authors present epidemiological data on influenza and its complications.
Material and method
The authors point out that a report has been prepared according to the "Core Model® EUnetHTA", addressing the clinical and non-clinical aspects.
What were the criteria for choosing this report?
Rows 136-139:
Because CVs among risk categories in individuals ≥18 years were not available, we applied a correction based on the relative risk (RR) of being vaccinated having at least one underlying chronic disease using the data of "Passi" [23] and "Passi d’argento" [24] surveillance systems.
How did you choose to apply the correction based on relative risk?
Row 184-188
,,The ethical analysis was referred to three principles: beneficence/non-maleficence (analysis of risk-benefit ratio; quality of life assessment); autonomy (respect of the individual’s freedom and responsibility; request for informed consent); justice and equity (cost-effectiveness ratio; prioritisation of health care policies; equal access to the technology".
Please specify which normative acts, criteria or approvals you used to present the data regarding the ethical analysis?
Rows 228-236
,,Relating to aTIV, our updated systematic review and meta-analysis [9] showed that aTIV was more immunogenic than eTIV versus both homologous and drifted strains; meta-analytic estimates reported that the magnitude of the antibody response was 26– 51% higher in subjects vaccinated with aTIV respect to vaccinee with eTIV. To note, that the advantage of aTIV vs TIVe was higher (+ 35%) regarding to the drifted A(H3N2) strains. Its absolute effectiveness was generally significantly different from zero, independently of setting and influenza season and it was generally highly effective [34.6% (95% CI: 2.0–66.0%; P<0.05) compared to the eTIV or eQIV in preventing laboratory- confirmed influenza".
The authors present the results of a meta-analysis performed by them showing the efficacy of aTIV vs TIVe for strain A (H3N2) and mention that adjuvanted vaccines may cause more adverse events compared to vaccines without adjuvant.
Can you briefly provide more information in this context?
And I think it's a mistake to write the result here: [34.6% (95% CI: 2.0–66.0%; P<0.05) -missing square bracket?
Rows 286-289
In the National Immunization Plan 2017-2019 [30] that is still in force in Italy influenza vaccination is recommended to individuals over the age of 64 years regardless of any risk situation.
Rows 292-295
In the Circular for 2020/2021 influenza season, instead, the indication for the elderly was more general: adjuvanted and non-adjuvanted vaccines, as well as high-dose and cell-based vaccines are all recommended for people aged ≥65 years, without any specific indication of use [16].
Rows 295-297
In the last Circulation for the 2021/2022 season, instead, adjuvated or high-dose QIV are reported as the most indicated vaccines in subjects ≥65 years of age [36].
Can you detail? Are these recommendations (rows 286-289, 292-295, 295-297) taken in the context of epidemiological analysis and influenza prevention programs for the vulnerable age group?
Rows 325-327
,,Eventually, considering the ethical dimension, collected data showed that aQIV induces an excellent antibody response against the four influenza-strains in the elderly; has a good safety profile; and is cost-effective".
What do you mean by "ethical dimension"?
Discussions
In the Discussion Chapter, the authors show the context of vaccine use and make recommendations and useful indications for vaccination campaigns for population groups.
Conclusions
In conclusion, the authors point out that there is a new vaccine (aQIV) available for immunizing the elderly.
Author Response

(The authors gave the same response as above.)

Round 2
Reviewer 1 Report
There are no comments and suggestions for authors.